# Inpatient Hospice Palliative Care Unit and Palliative Consultation Service Enhance Comprehensive Quality of Life Outcomes in Terminally Ill Cancer Patients: A Prospective Longitudinal Study

**DOI:** 10.3390/ijerph18178992

**Published:** 2021-08-26

**Authors:** Li-Fang Chang, Li-Fen Wu, Chi-Kang Lin, Ching-Liang Ho, Yu-Chun Hung, Hsueh-Hsing Pan

**Affiliations:** 1Department of Nursing, Tri-Service General Hospital, Taipei City 11490, Taiwan; fang_niki@mail.ndmctsgh.edu.tw (L.-F.C.); wulifen@mail.ndmctsgh.edu.tw (L.-F.W.); 2Graduate Institute of Medical Sciences, School of Nursing, National Defense Medical Center, Taipei City 11490, Taiwan; 3Department of Gynecology and Obstetrics, Tri-Service General Hospital, National Defense Medical Center, Taipei City 11490, Taiwan; kung568@mail.ndmctsgh.edu.tw; 4Division of Hematology and Oncology, Tri-Service General Hospital, National Defense Medical Center, Taipei City 11490, Taiwan; a2241@mail.ndmctsgh.edu.tw; 5Nursing Department, University of Kang Ning, Taipei City 11405, Taiwan; 6Department of Nursing, Tri-Service General Hospital, School of Nursing, National Defense Medical Center, Taipei City 11490, Taiwan

**Keywords:** terminally ill cancer patients, comprehensive quality of life outcomes, hospice palliative care, hospice palliative care unit, hospice palliative consultation service

## Abstract

This study aimed to explore the effectiveness of an inpatient hospice palliative care unit (PCU) and palliative consultation service (PCS) on comprehensive quality of life outcome (CoQoLo) among terminally ill cancer patients. This was a prospective longitudinal study. Terminally ill cancer patients who met the inclusion criteria and received PCU or PCS in a northern Taiwanese medical center were recruited. The CoQoLo Inventory was used to measure CoQoLo level pre- and seven days following hospice care between August 2018 and October 2019. A total of 90 patients completed the study. No significant differences were found in CoQoLo levels between the PCU and PCS groups pre- and seven days following care. However, the CoQoLo level of patients significantly improved seven days following care in both PCU and PCS groups, compared with pre-hospice care. Patients’ age, religious belief, marital status, closeness with family, palliative prognostic index (PPI), and symptom severity were significant concerning CoQoLo levels after adjusting for patients’ baseline characteristics. PCU and PCS showed no difference in CoQoLo levels, but both of them can improve CoQoLo among terminally ill cancer patients. These patients could receive PCU or PCS to achieve a good CoQoLo at the end-of-life stage.

## 1. Introduction

Cancer was still the second leading cause of death in the United States in 2018 [1]. Furthermore, a statistical survey estimated 1,806,590 new cancer cases and 606,520 related deaths in the United States in 2020 [2]. However, cancer has been ranked as the tenth cause of death in Taiwan for over 35 years [3]. To enhance the quality of life of terminally ill patients, hospice palliative care (HPC) has been advocated by the Ministry of Health and Welfare of Taiwan since 1996 [4], and the country’s legislative body passed the Hospice Palliative Care Act in 2000 [5]. Coinciding with these events, HPC utilization among terminal cancer patients in recent years has increased from 7% in 2000 to 61% in 2017 [6].

HPC refers to a patient-centered holistic care model that helps relieve pain, respect patients’ wills, and enhance quality of life, including the physical, psychological, social, and spiritual domains, among terminally ill patients and their family members [4,5]. At present, several types of HPC services are available in Taiwan, such as the inpatient hospice palliative care unit (PCU), inpatient hospice palliative consultation service (PCS), home-based hospice palliative care, and special hospice palliative care outpatient clinics [4]. PCU and PCS both provide HPC to inpatients. The difference between them is that PCU provides HPC in a special hospice palliative care unit, and PCS provides HPC in the general ward. HPC has been shown to decrease medical utilization and expenses [7,8,9,10,11], increase the chance of dying at home [12], raise the awareness of dying [13], and improve the rate of do-not-resuscitate signing and quality of end-of-life care [14]. The ultimate goal of HPC is to enable terminally ill patients to have a good death as well as good comprehensive quality of life outcome (CoQoLo). A good death for terminally ill patients largely means comfort, well-controlled pain and symptoms, explicit decision making, a sense of closure, being seen as a person, exhibiting respect for the patient’s wishes, awareness, preparation and acceptance of death, giving something to others, having their burdens minimized, religious and spiritual wellness, and having their relationships optimized [15,16,17]. Preparing for a good quality of dying and death is necessary in Chinese culture, as people expect those who remain to make efforts to address the final engagements of those who pass on. Previous studies have shown that the effectiveness of PCU with regard to care for terminally ill cancer patients is positively related to the quality of dying and death [18,19]. A cross-sectional study indicated that home-based hospice palliative care could achieve higher CoQoLo levels for cancer patients than palliative care units, as rated by bereaved family members [20]. However, these previous studies either did not explore the effectiveness of different HPC styles, used a cross-sectional design, or had family members rate the good death level. Few studies have used a prospective research method to examine the effectiveness between different HPC styles on CoQoLo, as rated by the patients themselves. Therefore, the objective of this study aimed to explore the effectiveness between PCU and PCS on CoQoLo among terminally ill cancer patients through the use of a prospective follow-up research design.

## 2. Methods

### 2.1. Study Design and Population

This study used a prospective longitudinal research design. Two groups, comprising terminally ill cancer patients diagnosed by physicians and those prepared to receive either PCU or PCS chosen by patients or families, were recruited from an approximately 1800-bed and 15-bed PCU medical center in northern Taiwan between August 2018 and October 2019. The inclusion criteria were as follows: had a life expectancy of less than six months, were over 20 years old, were conscious and alert, could communicate in Mandarin or Taiwanese, could understand the disease condition, were aware of the study’s purpose, and were willing to participate.

### 2.2. PCU and PCS

The process flow of PCU and PCS at the medical center is included below:

When terminally ill inpatients require HPC, they may choose either PCU or PCS. For PCU, patients would come from home or general wards and be transferred to PCU to receive HPC, whereas for PCS, patients receive HPC in general wards.

Before patients are transferred to receive HPC, the physicians or HPC nurses explain its concept to the patients and their family members. The patients start receiving PCU or PCS after confirming their willingness to participate and providing their informed consent.

A multidisciplinary team provides PCU or PCS, often composed of HPC specialists, HPC nurses, social workers, and a chaplain. The goal of both PCU and PCS is to assess the patients and their families’ conditions, including the physical, psychological, social, and spiritual domains, and offer suggestions regarding symptom control and psychosocial and family support.

The PCU team members provide care and assess patients’ and their families’ conditions directly. The PCS team visits the patient once or twice a week. During each visit, the teams evaluate the condition of the patients and their family members. The evaluation results are then reported and discussed during weekly HPC team meetings.

HPC team members hold a meeting once a week. They discuss the condition of cases including PCU, PCS, and home-based HPC, and then offer the patient’s future care plan to the patients and their families. PCU and PCS may include evaluating the possibilities of staying in the initial primary care facility, dispensing with home-based HPC, or discharge. For PCS, an evaluation of the possibilities of transferring to a PCU may be included.

PCU and PCS are terminated when the issues of the patients and their family members are resolved, or when the patient is transferred to a home-based HPC, is discharged, or has died. PCS is also terminated when patients are transferred to a PCU.

### 2.3. Measurements

#### 2.3.1. Patient Characteristics

Patient data collected included the patients’ age, sex, educational level, religious belief, marital status, and caregiver identity (spouse, child, or others). Furthermore, the data included whether they lived with others, their closeness with their family (measured using a Likert scale of 1–5, with 1 = no closeness at all and 5 = very close), economic source (self or others), and economic status (<20,000 NTD or ≥20,000 NTD). Moreover, the data collected included the number of times they have experienced deaths of family and friends (≤2 times or >2 times), perceived disease severity (Likert scale of 1–5, with 1 = not serious at all and 5 = very serious), duration of cancer diagnosis, palliative prognostic index (PPI), and symptom severity.

#### 2.3.2. Palliative Prognostic Index (PPI)

The PPI is used to predict survival in terminal cancer patients [21]. This is determined using the variables of performance status (0, 2.5, and 4 points), oral intake (0, 1, and 2.5 points), edema (0 and 1 points), dyspnea at rest (0 and 3.5 points), and delirium (0 and 4 points). Total possible PPI scores range from 0 to 15. A lower PPI score indicates a longer survival time. A PPI score of more than six indicates a predicted survival time of fewer than three weeks, whereas a PPI score of more than four indicates a predicted survival time of fewer than six weeks. The PPI has been shown to have high sensitivity and specificity to predict survival time in terminally ill cancer patients [21].

#### 2.3.3. Memorial Symptom Assessment Scale Short Form (MSAS-SF)T

The memorial symptom assessment scale (MSAS) was initially developed by Portenoy et al. [22] and revised to a short form version by Chan et al. [23]. The MSAS-SF has been translated into Chinese and was demonstrated to have good validity and reliability [24]. The scale is a patient-reported symptom severity scale that assesses 28 physical and four psychological symptoms exhibited by the respondent during the past week. Physical symptoms include lack of energy, pain, lack of appetite, drowsiness, constipation, dry mouth, nausea, vomiting, change in taste, weight loss, feeling bloated, dizziness, shortness of breath, difficulty sleeping, cough, numbness and tingling, difficulty concentrating, urination, sweats, itching, not looking like self, changes in the skin, lack of sexual interest, swelling of arms/legs, difficulty swallowing, diarrhea, hair loss, and mouth sores. In contrast, psychological symptoms include feeling sad, worrying, feeling irritable, and feeling nervous. Each symptom’s severity is rated on a five-point scale, with 0 representing “no symptom” and 4 “very severe”. The total score of symptom severity ranges from 0 to 128. A higher score represents more severe symptoms [24]. In the present study, Cronbach’s α was 0.724.

#### 2.3.4. Comprehensive Quality of Life Outcomes (CoQoLo) Inventory

The CoQoLo inventory was developed by Miyashita et al. and is used to assess one’s CoQoLo level based on the concept of good death for patients with advanced cancer [25]. This is a self-evaluated scale comprising 28 items and 10 subscales (i.e., physical and psychological comfort, staying in a favorite place, maintaining hope and feeling pleasure, good relationships with medical staff, not being a burden to others, good relationships with family, independence, environmental comfort, being respected as an individual, and having a fulfilling life). Each item is rated on a seven-point Likert scale, with 1 representing “completely disagree” and 7, “completely agree.” The total scores of CoQoLo level range from 28 to 196. A higher score represents a good CoQoLo level. The good validity and reliability of this inventory have been demonstrated in a prior study [25]. Cronbach’s α was 0.845 for 105 patients with terminally ill cancer in this study.

### 2.4. Study Process

This study was approved by the institutional review board of the Tri-Service General Hospital (TSGHIRB No.:2-107-05-028). Terminally ill cancer patients who fulfilled the inclusion criteria were prepared to receive either PCU or PCS and were provided with information on the study protocol. Before starting the experiment, the researcher explained the study’s objectives and methods to the participants in a designated meeting room. The same researcher collected data via questionnaires before the PCU or PCS was provided and seven days following it after each participant signed the informed consent. The questionnaires were undisclosed, and the collected information was to be considered confidential. Participants spent approximately 15 to 20 min filling in the questionnaires and they received a gift card after filling out the questionnaires. Participants were allowed to stop the study at any time.

### 2.5. Statistical Analysis

Data were encoded using the spreadsheet software Microsoft Excel and analyzed using the software IBM SPSS Statistics for Windows, version 23.0 (IBM Corp., Armonk, NY, USA). Categorical variables were stated as frequencies and percentages, and continuous variables were stated as means and standard deviations (SDs). Baseline heterogeneity between the PCU and PCS groups was analyzed using a *t*-test and chi-square test. Fisher’s exact test was also used to adjust the cell count to less than five. The difference in CoQoLo level before and seven days following hospice care was analyzed using a paired *t*-test. A generalized estimating equation (GEE) was used to analyze the data, including repeat measurements, which were often collected at different time points for each individual. GEE could be used to manage intra-person dependency to explore the independent impact factors on specific outcomes. Moreover, it could adjust potential confounding factors to present the effect of the used variable. Therefore, we used GEE to determine the predictors of CoQoLo levels of terminally ill cancer patients before and seven days following PCU and PCS. A statistically significant *p*-value was considered as a value lower than 0.05.

## 3. Results

Data were collected from 105 terminally ill cancer patients, of which 70 patients were prepared to receive PCU, and 35 patients were prepared to receive PCS. A total of 64 patients received PCU, and 26 patients received PCS and completed the questionnaire seven days following hospice care. A flowchart illustrating the flow of the study is shown in Figure 1. The baseline characteristics of the terminal cancer patients who received PCU and those who received PCS are shown in Table 1. No significant differences in baseline characteristics were noticed between the two groups.

There were no statistically significant differences in the mean values of the CoQoLo level between the PCU and PCS groups before (125.6 ± 20.2 vs. 122.1 ± 20.3, *p* = 0.403) and seven days following care (136.2 ± 14.6 vs. 132.2 ± 16.4, *p* = 0.211). However, the CoQoLo levels seven days after care significantly improved not only in the PCU group but also in the PCS group compared with the pre-care levels (*p* < 0.001; see details in Table 2).

We used the GEE model of the CoQoLo level to test each baseline characteristic variable of the terminally ill cancer patients before and seven days following PCU and PCS. The CoQoLo level of the terminally ill cancer patients was shown to have improved following PCU and PCS (β = 5.3, *p* = 0.020). The patients’ age (β = 0.3, *p* = 0.041), presence of religious beliefs (β = 8.7, *p* = 0.013), marital status (β = −7.8, *p* = 0.029), closeness with family (β = 2.8, *p* = 0.011), PPI (β = 1.5, *p* = 0.011), and symptom severity (β = −0.4, *p* < 0.001) were statistically significant, with CoQoLo levels being exhibited after adjusting for the baseline characteristics of patients (see details in Table 3).

## 4. Discussion

This study is one of the first to discuss the effect of PCU and PCS on CoQoLo among terminally ill cancer patients through the use of patient-reported outcomes and a prospective longitudinal design. We found that there were no significant differences in the levels of CoQoLo between the PCU and PCS groups. A prior study showed that patients admitted to the PCU had a worse performance status and severe symptoms and psychosocial problems compared with those consulted by the PCS [26]. A survey also indicated that family members of patients who died on the PCU were more likely to report that patients had better end-of-life medical care and emotional support than those who received PCS [27]. However, the baseline characteristics of this study showed no difference in perceived disease severity, PPI, and symptom severity in the PCU and PCS groups. In our medical center, both PCU and PCS provide patients with a multidisciplinary team composed of different professionals, including physicians, hospice specialist nurses, psychologists, social workers, pharmacists, and chaplains. Additionally, PCU and PCS provide holistic patient-centered care to the patients at the end stage as needed, and assist in medical decision-making processes related to advanced care planning. In Taiwan, our government has been actively promoting PCS due to the limited number of beds with regard to PCU. Therefore, we expected that this policy could lead to more terminally ill patients receiving PCU or PCS in order to achieve a good CoQoLo in the end stage.

This study indicated that the CoQoLo level of terminally ill cancer patients significantly improved not only among those who received PCU seven days following the care but also among those who received PCS. This finding was similar to several previous studies [18,28,29]. Terminally ill cancer patients who had received PCU or PCS had significantly less medical utilization and expenses, lower aggressive medical treatments, and a higher quality of dying, as compared to those who never received PCU or PCS [7,28,30,31,32]. CoQoLo is assessed by patients themselves and means the patient’s physical and psychological comfort, staying in a favorite place, maintaining hope and feeling pleasure, good relationships with medical staff, not being a burden to others, good relationships with family, independence, environmental comfort, being respected as an individual, and having a fulfilling life to achieve a good death for patients with advanced cancer [25]. A good death is usually assessed by family members after the patient dies. It is one of the primary outcomes of end-of-life care by the patient and the family and highly individualistic, changeable over time, and based on perspective and experience [33]. Physical comfort, being pain-free, and psychosocial and spiritual peace should be considered in addition to the crucial components of preparation for a good CoQoLo as well as good death [34]. Therefore, the goal of PCU and PCS is not to extend life or accelerate death but to increase the quality of life and promote dignified death [35].

In this study, older terminally ill patients had higher CoQoLo levels, which is not consistent with prior findings [36,37]. Older patients reported lower attainment of feelings of hope and pleasure in their relationship with their families, possibly resulting in loneliness and seclusion [38]. Moreover, elderly patients with terminal cancer in Taiwan generally do not fully understand their illness condition even before death. In Taiwanese culture, it is not an ordinary practice to disclose the entire truth regarding an illness to patients based on non-maleficence [36]. However, a study has revealed that older terminally ill cancer patients are more likely to show death preparedness [39]. Older patients usually have positive attitudes toward death, exhibit ambiguous emotions or attitudes in the confrontation with death, and are more likely to be at peace [40,41,42].

Our research found that having religious beliefs might affect the level of CoQoLo among terminally ill cancer patients. Religious beliefs are positively related to the perception of preparation for death and spiritual well-being and are negatively associated with physical suffering [43]. It is an essential element in the practices and aspects that lead to a good CoQoLo [44]. Patients who are religious receive more spiritual support via their religion to overcome any fear of death. Religious belief serves as a spiritual inspiration and becomes more important for terminally ill patients as they feel the need for natural or supernatural assistance near their end of life [43]. Furthermore, terminally ill cancer patients with religious beliefs seem to have little fear of dying, suggesting that both religious beliefs and practices could reduce anxiety related to death [45].

Our findings revealed that terminally ill patients closer to their families had a higher level of CoQoLo. Studies have shown that having a close family is a key element of a good death among terminally ill patients [44,46]. Improved death preparedness is related to higher social support in terminally ill cancer patients [39]. Close family members endeavor for balance and well-being to attend their affected relative at the end of their life. They help them find meaning and strength by helping, and contribute to their regaining balance and harmony in the terminal stage [47]. Additionally, close family members can support terminally ill patients to be able to say goodbye, pronounce euthanasia in the case of intolerable suffering, and help accomplish a good death as well as CoQoLo [48].

This study indicated that having higher PPI scores meant a shorter survival time, relating to higher CoQoLo levels among terminally ill cancer patients. A study has shown that short or long survival time does not affect the accomplishment of CoQoLo in patients with terminal cancer [49]. Patients who received hospice care for 22–84 days have been reported to exhibit a significantly higher quality of dying than those who received hospice care for 3–21 days [19]. Such a short course of time can pose a challenge in providing hospice care; however, with the hospice care team members’ endeavor, CoQoLo can be achieved within a short survival time. Therefore, patients with terminal cancer can experience a good CoQoLo as well as good death, even with short survival times [49].

Patients with lower symptom severity in this study exhibited higher CoQoLo levels. Patients usually become sicker, and physical symptoms inevitably increase close to the end of life [49]. A previous study showed that terminally ill cancer patients with fewer distressing symptoms tend to report death preparedness [39]. Notably, severe symptom distress possibly affects prognostic awareness and poses accompanying challenges in achieving a good CoQoLo among these patients [50]. Being free from pain or other distressing symptoms is key to achieving CoQoLo in terminally ill cancer patients [15,16,17].

This study was a prospective longitudinal design using patient-reported CoQoLo levels. However, it also has some limitations. First, the patients investigated had clear consciousness and were a small sample size from a single medical center. Hence, generalizing this study’s conclusions to all terminally ill cancer patients may be inappropriate. Second, we did not enroll patients who had not received PCU or PCS because patients did not know their disease condition or their doctor had not yet determined patients to be in their terminal stage. Therefore, the level of CoQoLo among these people was unknown. Finally, the current study only followed up on the CoQoLo level of terminally ill cancer patients before and seven days following care. Consequently, the long-term effects of PCU and PCS on CoQoLo levels are unknown.

Based on the limitations of this study, future research directions are suggested. First, studies using different medical centers or hospitals and large sample sizes are needed to increase the findings’ applicability and generalizability. Second, a prospective cohort study that compares terminally ill cancer patients, either involving or not involving PCU and PCS, is recommended to understand the levels of CoQoLo among these groups. Finally, continuous follow-up on the CoQoLo level of terminally ill cancer patients is essential to confirm the effects of PCU and PCS.

## 5. Conclusions

This study supported that there were no differences in the levels of CoQoLo between the PCU and PCS groups. The CoQoLo levels of terminally ill cancer patients significantly improved not only in the PCU group but also in the PCS group over time following care. These findings highlighted the substantial benefits of providing inpatient palliative care, PCU and PCS, and supported further promotion more widely in terminally ill cancer patients to enhance a good CoQoLo at the end-stage of their life.

## Figures and Tables

**Figure 1 ijerph-18-08992-f001:**
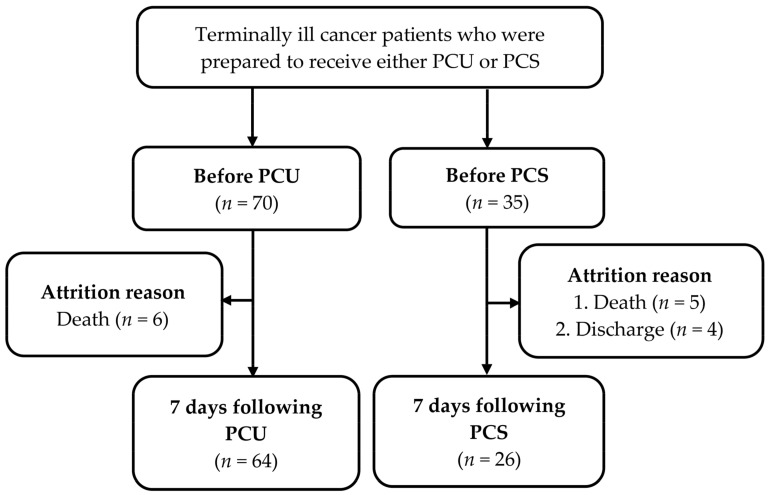
The study flowchart. PCU: palliative care unit; PCS: palliative consultation service.

**Table 1 ijerph-18-08992-t001:** Baseline characteristics of terminally ill cancer patients receiving inpatient hospice palliative care unit (PCU; *n* = 70) and palliative consultation service (PCS; *n* = 35).

Variable	PCU	PCS	
	(*n* = 70)	(*n* = 35)	*p* Value
	Mean ± SD/*n* (%)	Mean ± SD/*n* (%)	
Age	67.6 ± 12.0	65.1 ± 14.6	0.348
Sex			0.779
Male	42 (60.0)	20 (57.1)	
Female	28 (40.0)	15 (42.9)	
Educational level			0.162
High school and below	44 (62.9)	17 (48.6)	
Specialist or above	26 (37.1)	18 (51.4)	
Religious belief			0.290
No	23 (32.9)	8 (22.9)	
Yes	47 (67.1)	27 (77.1)	
Marital status			0.238
Single	20 (28.6)	14 (40.0)	
Married	50 (71.4)	21 (60.0)	
Child			1.000 ^a^
No	7 (10.0)	4 (11.4)	
Yes	63 (90.0)	31 (88.6)	
Living with others			0.330 ^a^
No	2 (2.9)	3 (8.6)	
Yes	68 (97.1)	32 (91.4)	
Caregiver identity			0.336 ^a^
None	1 (1.4)	3 (8.6)	
Spouse	32 (45.7)	17 (48.6)	
Child	22 (31.4)	9 (25.7)	
Others	15 (21.4)	6 (17.1)	
Closeness with family	3.0 ± 1.3	2.9 ± 1.3	0.833
Economic sources			0.549
Self	50 (71.4)	23 (65.7)	
Others	20 (28.6)	12 (34.3)	
Economic status			0.479
<20,000 NTD	29 (41.4)	12 (34.3)	
≥20,000 NTD	41 (58.6)	23 (65.7)	
Experienced deaths of family and friends			0.263
≤2 times	15 (21.4)	11 (31.4)	
>3 times	55 (78.6)	24 (68.6)	
Perceived disease severity	3.3 ± 0.6	3.4 ± 0.8	0.471
Duration of cancer diagnosis (years)	4.2 ± 2.4	3.8 ± 2.6	0.828
Palliative prognostic index	4.7 ± 3.0	4.4 ± 2.7	0.640
Symptom severity	29.3 ± 11.2	27.7 ± 13.2	0.521

PCU: palliative care unit; PCS: palliative consultation service; SD: standardized deviation. ^a^ Testing by Fisher’s exact test, Wilcoxon test, or Kruskal–Wallis test, respectively.

**Table 2 ijerph-18-08992-t002:** CoQoLo level over time following inpatient hospice palliative care unit (PCU) and palliative consultation service (PCS) among terminally ill cancer patients.

Variables	PCU	PCS	
	Mean ± SD	Mean ± SD	*p* Value
CoQoLo level			
Pre-care (70 vs. 35)	125.6 ± 20.2	122.1 ± 20.3	0.403
Seven days following care (64 vs. 26)	136.2 ± 14.6	132.2 ± 16.4	0.211
Mean difference pre- and seven days following care	10.6 ± 14.7	10.1 ± 16.7	0.886
Paired *t*-test	*p* < 0.001	*p* < 0.001	

CoQoLo: comprehensive quality of life outcomes; PCU: palliative care unit; PCS: palliative consultation service; SD: standardized deviation.

**Table 3 ijerph-18-08992-t003:** Predictors of CoQoLo level of terminally ill cancer patients pre- and seven days following PCU and PCS (*n* = 90).

Variable.	Crude β (95% CI)	*p* Value	Adjusted β (95% CI)	*p* Value
Time * hospice palliative style				
Pre-care				
PCU/PCS	Reference		Reference	
Seven days following care				
PCU/PCS	1.7 (−3.7–7.3)	0.838	3.1 (−2.3–8.5)	0.257
Time				
Pre-care	Reference		Reference	
Seven days following care	11.2 (7.0–15.4)	<0.001	5.3 (0.8–9.8)	0.020
Hospice palliative style				
PCU	1.2 (−6.6–9.1)	0.157	0.7 (−6.2–7.5)	0.850
PCS	Reference		Reference	
Age	0.2 (−0.002–0.5)	0.052	0.3 (0.01–0.5)	0.041
Sex				
Male	Reference		Reference	
Female	2.3 (−3.5–8.1)	0.443	0.5 (−4.8–5.8)	0.858
Educational level				
High school and below	Reference		Reference	
Specialized or above	−1.6 (−7.8–4.6)	0.614	4.5 (−1.1–10.1)	0.117
Religious belief				
No	Reference		Reference	
Yes	6.4 (−0.3–13.0)	0.061	8.7 (1.8–15.5)	0.013
Marital status				
Single	Reference		Reference	
Married	−5.3 (−11.4–0.8)	0.089	−7.8 (−14.9–−0.8)	0.029
Child				
No	Reference		Reference	
Yes	2.2 (5.0–−7.6)	0.661	−4.8 (−17.2–7.5)	0.442
Living with others				
No	Reference		Reference	
Yes	−3.6 (−14.1–6.9)	0.501	−15.7 (−34.1–2.7)	0.095
Caregiver				
No	Reference		Reference	
Spouse	−4.4 (−20.5–11.6)	0.589	10.7 (−13.2–34.6)	0.379
Child	−1.6 (−17.8–14.6)	0.038	10.2 (−13.8–34.1)	0.405
Others	−5.5 (−21.6–10.7)	0.507	2.1 (−20.7–24.9)	0.858
Closeness with family	2.5 (0.1–4.8)	0.041	2.8 (0.6–5.0)	0.011
Economic source				
Self	Reference		Reference	
Others	−0.4 (−6.0–5.2)	0.881	1.2 (−4.2–6.6)	0.659
Economic status				
<20,000 NTD	Reference		Reference	
≥20,000 NTD	−1.7 (−8.0–4.6)	0.598	−5.0 (−10.2–0.2)	0.061
Experienced deaths of family and friends				
<3 times	Reference		Reference	
≥3 times	5.3 (−1.4–12.0)	0.119	2.0 (−5.1–9.0)	0.581
Perceived disease severity	0.7 (−4.5–5.9)	0.797	0.02 (−4.5–4.5)	0.995
Duration of cancer diagnosis (years)	−0.2 (−1.2–0.9)	0.736	0.2 (−0.8–1.1)	0.733
Palliative prognostic index	1.5 (0.6–2.4)	0.002	1.5 (0.6–2.4)	0.001
Symptom severity	−0.6 (−0.8–−0.4)	<0.001	−0.4 (−0.6–−0.2)	<0.001

CoQoLo: comprehensive quality of life outcomes; PCU: palliative care unit; PCS: palliative consultation service. *: interaction.

## Data Availability

The data presented in this study are available on request from the corresponding author. The data are not publicly available due to privacy.

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
