# Peer review of "Inpatient Hospice Palliative Care Unit and Palliative Consultation Service Enhance Comprehensive Quality of Life Outcomes in Terminally Ill Cancer Patients: A Prospective Longitudinal Study"

_ijerph, 2021, doi:10.3390/ijerph18178992_

Round 1

Reviewer 1 Report

Thank you for the interesting manuscript. The topic may be interesting to a wide audience but in my opinion a major revision is needed before the manuscript may be accepted for publication.

I recommend to perform a major revision of the paper and to address some major concerns:

1) You state that the CoQoLo was used to measure good death level. This is not correct from my point of view. The CoQoLo is used to measure QOL outcomes and not "good death level". I am a bit puzzled if you are talking about the good death inventory here? As reader I am lost in your description and have a lot of questions. To me it looks like a mix of CoLoQo and GDI. Please describe this in detail to enable understanding. Therefore you should revise the paper and make it clear which tool is used and what you aim to measure and investigate. 

2) I miss a clear definition of good death as used in your manuscript.

3) I think your use of the term good death level is puzzling and in part irritating for the reader. What do you mean by that term? Needs a clear definition for the use in your paper.

4) Methods: How were participants included in both groups? Randomisation or participants own choice? How can you use the CoLoQo to assess good death levels?

5) You write that preparing for a good death is necessary for the Chinese culture. This should be explained in m ore detail for the readers of an international journal. References for that are needed in a scientific paper. 

6) As some of my problems to understand your paper may be based in the use of English language I recommend English proof-reading.

I hope that my comments will help you to revise and improve your paper. Good luck with a revision of your manuscript.

Author Response

Reviewer1#

Thank you for the interesting manuscript. The topic may be interesting to a wide audience but in my opinion a major revision is needed before the manuscript may be accepted for publication.

I recommend to perform a major revision of the paper and to address some major concerns:

  • You state that the CoQoLo was used to measure good death level. This is not correct from my point of view. The CoQoLo is used to measure QOL outcomes and not "good death level". I am a bit puzzled if you are talking about the good death inventory here? As reader I am lost in your description and have a lot of questions. To me it looks like a mix of CoLoQo and GDI. Please describe this in detail to enable understanding. Therefore you should revise the paper and make it clear which tool is used and what you aim to measure and investigate. 

Authors’ response:

Thank you very much for the valuable comments. We have revised in the Introduction and Method sections and all revised manuscript using “Track Changes”.

  • I miss a clear definition of good death as used in your manuscript.

Authors’ response:

Thank you very much. We have revised a clear definition of good death as well as good comprehensive quality of life outcome (CoQoLo) in the Introduction using “Track Changes”.

“A good CoQoLo as well as good death for terminally ill patients largely means comfort, well-controlled pain and symptoms, explicit decision-making, a sense of closure, being seen as a person, exhibiting respect for the patient's wishes, awareness, preparation and acceptance for death, giving something to others, having their burdens minimized, religious and spiritual wellness, and having their relationships optimized [15-17]”

  • I think your use of the term good death level is puzzling and in part irritating for the reader. What do you mean by that term? Needs a clear definition for the use in your paper.

Authors’ response:

Thank you very much. We have revised the term good death to CoQoLo. A clear definition of CoQoLo was showed in the revised manuscript, especially in Introduction using “Track Changes”.

“A good CoQoLo as well as good death for terminally ill patients largely means comfort, well-controlled pain and symptoms, explicit decision-making, a sense of closure, being seen as a person, exhibiting respect for the patient's wishes, awareness, preparation and acceptance for death, giving something to others, having their burdens minimized, religious and spiritual wellness, and having their relationships optimized [15-17]”

  • Methods: How were participants included in both groups? Randomisation or participants own choice? How can you use the CoLoQo to assess good death levels?

Authors’ response:

Thank you very much for the comment.Two groups, comprising terminally ill cancer patients diagnosed by physicians and those prepared to receive either PCU or PCS choosed by patients or families, were recruited from an approximately 1,800-bed and 15-bed PCU medical center in northern Taiwan between August 2018 and October 2019.” In addition, we have revised the term good death to CoQoLo.

  • You write that preparing for a good death is necessary for the Chinese culture. This should be explained in more detail for the readers of an international journal. References for that are needed in a scientific paper. 

Authors’ response:

Thank you very much for the suggestions. We have explained in more detail in the Introduction section using “Track changes”. In addition, references have cited in that paragraph.

Preparing for a good CoQoLo is necessary in Chinese culture as people expect those who remain to make efforts to address the final engagements of those who pass on. Previous studies have shown that the effectiveness of PCU with regards to care for terminally ill cancer patients is positively related to the quality of dying and death [18, 19].”

  • As some of my problems to understand your paper may be based in the use of English language I recommend English proof-reading.

Authors’ response:

Thank you very much. We have edited by English expert. 

Reviewer 2 Report

There are minor errors in the language. Would be helpful if we could know from the terminally ill patient what they would consider is a good death...

Does symptom control, etc mean that death is good?

Author Response

Reviewer 2#

There are minor errors in the language. Would be helpful if we could know from the terminally ill patient what they would consider is a good death...

Authors’ response:

Thank you very much. We have edited by English expert. The certification is following. We have explained in more detail in the Introduction section using “Track changes”.

“The ultimate goal of HPC is to enable terminally ill patients to implement a good comprehensive quality of life outcome (CoQoLo). A good CoQoLo as well as good death for terminally ill patients largely means comfort, well-controlled pain and symptoms, explicit decision-making, a sense of closure, being seen as a person, exhibiting respect for the patient's wishes, awareness, preparation and acceptance for death, giving something to others, having their burdens minimized, religious and spiritual wellness, and having their relationships optimized [15-17].”

Does symptom control, etc mean that death is good?

Authors’ response:

Thank you very much for the comment. We have revised a clear definition of good death as well as good comprehensive quality of life outcome (CoQoLo) in the Introduction using “Track Changes”.

“A good CoQoLo as well as good death for terminally ill patients largely means comfort, well-controlled pain and symptoms, explicit decision-making, a sense of closure, being seen as a person, exhibiting respect for the patient's wishes, awareness, preparation and acceptance for death, giving something to others, having their burdens minimized, religious and spiritual wellness, and having their relationships optimized [15-17]”

Round 2

Reviewer 1 Report

Thank you for the revision of your paper and clarification of the used terms! 

I think the paper may now be published but I would like to see a short discussion of the term good death versus good CoQoLo. This could be interesting to all readers. 

Author Response

Reviewer1#

I think the paper may now be published but I would like to see a short discussion of the term good death versus good CoQoLo. This could be interesting to all readers.

Authors’ response:

Thank you very much for the valuable comments. We have revised in the Discussion section and all revised manuscript was used “Track Changes” in page 8.

“CoQoLo is assessed by patients themselves and means the patient’s physical and psychological comfort, staying in a favorite place, maintaining hope and feeling pleasure, good relationships with medical staff, not being a burden to others, good relationships with family, independence, environmental comfort, being respected as an individual, and having a fulfilling life to achieve a good death for patients with advanced cancer [25]. A good death is usually assessed by family members after the patient dies. It is one of the primary outcomes of end-of-life care by the patient and the family and highly individualistic, changeable over time, and based on perspective and experience [33]”
